# Biological Activities of Lactic Acid Bacteria Isolated from Chinese Traditional Cheese and the Application in Antioxidant Foods

**DOI:** 10.3390/microorganisms13122743

**Published:** 2025-12-02

**Authors:** Xiangdi Lou, Liping Lin, Wenwu Zhu, Xiaochen Zhang, Jianhua Xiong, Yanyan Gao

**Affiliations:** 1College of Food Science and Engineering, Jiangxi Agricultural University, Nanchang 330045, China; 2Jiangsu Coastal Area Institute of Agricultural Science, Yancheng 224002, China

**Keywords:** lactic acid bacteria, antioxidant, PCA, EW-TOPSIS, milk fermentation

## Abstract

Lactic acid bacteria (LAB) are known for their health benefits, which are isolated frequently from various fermented foods. This study aimed to evaluate the safety, metabolite production, antioxidant, and antibacterial activity of LAB isolated from Chinese traditional cheese and their application in fermented dairy products. According to 16S rRNA sequence analysis, seven LAB strains were identified as *Lactococcus lactis* W3A and W3C, *Streptococcus lutetiensis* W3B, *Enterococcus durans* W3D, *Leuconostoc mesenteroides* W3E, *Enterococcus lactis* W3F, and *Leuconostoc lactis* W3J, respectively. Hemolysis and antibiotic sensitivity tests showed that all strains were safe for consumers. Meanwhile, all strains showed a good ability to produce exopolysaccharide (330.57–1097.10 mg L^−1^) and organic acid (31.83–65.43 g L^−1^). The antioxidant assay indicated that seven LAB strains could effectively scavenge DPPH (37.72–53.13%), ABTS^+^ (95.86–98.67%), ·OH (86.06–95.26%), ·O_2_^−^ (11.88–31.30%) free radicals, and chelate ferrous ion (23.98–54.70%) and could reduce ferric ions (35.56–273.14 μmol L^−1^). In addition, they also displayed different antibacterial activity (10–30 mm inhibition zone) against all 13 foodborne pathogenic bacteria. Finally, the two best strains (*Leu. lactis* W3J and *E. lactis* W3F) were selected with PCA and EW-TOPSIS methods and used to ferment goat milk. Fermented samples provided a stronger antioxidant ability than the unfermented goat milk. The results suggested that these two LAB strains had potential applications in antioxidant foods.

## 1. Introduction

Probiotics are “live microorganisms, which when administered in adequate amounts confer a health benefit on the host” [1]. As generally recognized as safe (GRAS) microorganisms by the Food and Drug Administration (FDA), lactic acid bacteria (LAB) are the most widely used probiotics as starter cultures in a variety of functional foods to promote health [2]. More than 62 genera of LAB have been developed as starters for food fermentation [3]. Members of *Lactococcus*, *Lactobacillus*, *Leuconostoc*, *Streptococcus*, *Pediococcus*, and *Enterococcus* are recognized as the most important microorganisms in the dairy industry, and frequently are isolated from traditional fermented products [4]. Typical LAB strains naturally ferment carbohydrates to produce organic acids (mainly lactic acid), enzymes, and bacteriocins to combat microbial contaminations in food processing and storage [5]. Some LAB strains can also produce exopolysaccharide (EPS) or other active metabolites to improve the functional properties, nutritional qualities, and health benefits of certain foods [6].

Antioxidant activity is another prominent functional property of LAB. Free radicals are substances produced generally by the normal physiological metabolism in the human body. If free radicals are not removed in time, their accumulation may lead to aging, degenerative disease, and cancer [7]. The antioxidant activity of LAB has been confirmed by various methods. They inhibit occurrence and progression of aging and disease by free radical scavenging, reducing capacity and metal ion chelation. For example, Ramalho et al. [8] reported that *Lactococcus* from mozzarella cheese could effectively scavenge 2,2-diphenyl-1-picrylhydrazyl (DPPH), 2,20-azino-bis (3-ethylbenzothiazoline-6-sulfonic acid) (ABTS) free radicals, as well as reduced ferric ions. Kim et al. [9] demonstrated that 15 LAB strains from fermented foods including *Leuconostoc* and *Lactobacillus* decreased the levels of DPPH and ABTS free radicals. Nevertheless, different species of bacteria exhibited distinct activities in the different antioxidant assays, and even identical bacterial species varied antioxidant capacities. Thus, it is necessary to estimate the comprehensive antioxidant performance of LAB through various methods.

LAB, as novel natural preservatives, exhibit a good broad-spectrum antibacterial activity, including against *Escherichia coli*, *Salmonella enterica*, *Staphylococcus aureus*, *Bacillus cereus*, *Vibrio parahaemolyticus*, and *Listeria monocytogenes* [10,11]. Meanwhile, LAB may also prevent the colonization of pathogenic bacteria in the host by competing for nutrients or adhesion sites to stabilize the beneficial microbial population [12]. The antibacterial properties of LAB endow their role in food preservative and nutritional enhancement. Especially in traditional fermented foods, such as yogurt, cheese, and pickles, epiphytic LAB directly hindered the proliferation and invasion of other harmful bacteria, thereby reducing the use of chemical preservatives [12]. Therefore, the assessment of LAB on antibacterial activity is critical for the safety of fermented foods.

Rubing cheese is a traditional Chinese Protected Designation of Origin (PDO) cheese produced for more than 600 years in China’s southwest. Next generation sequencing results revealed that they retained abundant and excellent LAB resources such as *Lactococcus* and *Leuconostoc* during the long domestication process of traditional fermentation [13]. However, there are few reports about the isolation and characterization of endogenous LAB. In this study, seven indigenous LAB strains from rubing cheese were isolated and identified based on phylogenetic and molecular analysis. Then the biological properties of seven LAB strains were evaluated for the safety, metabolite production, and antioxidant and antibacterial activity. Finally, the two best strains were selected to further investigate the application in antioxidant foods.

## 2. Materials and Methods

### 2.1. Materials and Reagents

Pathogenic indicator strains including *Escherichia coli* ATCC 35218, *Shigella flexneri* CMCC 51572, *B. cereus* ATCC 14579, *Pseudomonas aeruginosa* ATCC 27853, *Staphylococcus aureus* ATCC 29213, *Streptococcus pyogenes* ATCC 19615, *Klebsiella pneumoniae* ATCC 4352, *Vibrio parahaemolyticus* ATCC 17802, *Aeromonas hydrophila* 3CA, *Aeromonas veronii* 5SB, *Listeria monocytogenes* ATCC 19115, *Salmonella enterica* serovar Paratyphi (*S.* Paratyphi) CMCC 50094, and *Salmonella enterica* serovar Enteritidis (*S.* Enteritidis) ATCC BAA708 were obtained from Shanghai Bioresource Collection Center (Shanghai, China) and stored at −80 °C. Columbia blood agar plates and LB agar were purchased from Hopebio Biotechnology Co., Ltd. (Qingdao, China). All antibiotic discs were purchased from Bkmam Biotechnology Co., Ltd. (Changde, China). DPPH, ABTS, and tripyridyltriazine (TPTZ) were purchased at Sigma-Aldrich Co., Ltd. (Beijing, China). Hydrogen peroxide (H_2_O_2_), trichloroacetic acid, and other reagents were purchased from Macklin Biotechnology Co., Ltd. (Shanghai, China). Rubing cheese was acquired from the farm workshops (Dali, China). Fresh goat milk was provided by Ruikang Dairy Co., Ltd. (Nanchang, China).

### 2.2. Isolation and Identification of LAB Strains

First, 10 g of rubing cheese was homogenized, diluted, and plated on modified de Man–Rogosa–Sharpe (mMRS-C) agar containing 0.5% calcium carbonate and 2% sucrose instead of glucose at pH 6.1 and incubated at 37 °C for 3 days under aerobic conditions. Characteristic colonies were selected, purified, and preliminarily identified via Gram staining analysis. Seven isolated strains were further identified by 16S rRNA amplification and sequencing (27F primer sequence: 5′-GTTTGATCMTGGCTCAG-3′; 1492R primer sequence: 5′-TACGGYTACCTTGTTACGACTT-3′) [14]. Identification of strains was carried out by sequence comparisons with reference sequences in GenBank using the BLAST software (version 2.17.0+). A neighbor-joining phylogenetic tree was constructed using MEGA 7.0 software to further evaluate the evolutionary relationship between the selected LAB strains. Nucleotide sequences of 7 LAB strains (accession numbers PQ443773–PQ443779) were deposited in GenBank database.

### 2.3. Safety

#### 2.3.1. Hemolytic Assay

All the LAB strains were spot inoculated on Columbia blood agar plates, and then the plates were examined for zones of hemolysis after incubation at 37 °C for 24 h [15]. *Bacillus cereus* ATCC 14579 was used as the positive control.

#### 2.3.2. Antibiotic Susceptibility

According to the standards of the Clinical and Laboratory Standards Institute (CLSI) guidelines, the antibiotic susceptibility of the LAB strains was assessed using the antibiotic disc diffusion method [16]. First, 100 μL of overnight LAB cultures was spread on MRS plates, and then antibiotic discs were placed on the plates. Antibiotics tested included penicillin (10 μg), piperacillin (100 μg), ampicillin (10 μg), cefalexin (30 μg), cefazolin (30 μg), cefoperazone (30 μg), amikacin (30 μg), tetracycline (30 μg), minocycline (30 μg), gentamicin (10 μg), streptomycin (10 μg), vancomycin (30 μg), and doxycycline (30 μg). After 24 h incubation at 37 °C, the diameters of inhibition zone (mm) were investigated.

### 2.4. The Preparation of Cell-Free Supernatant (CFS) of LAB

The LAB strain stored at −80 °C was incubated to mMRS medium (containing 2% sucrose instead of glucose) plates for activation. For each LAB strain, single colony was picked up from the plate and inoculated into 3 mL of fresh mMRS medium to culture 24 h at 37 °C. Then 1 mL of bacterial culture was transferred into 100 mL of fresh mMRS. After incubation for 36 h at 37 °C, cell-free supernatants (CFS) of LAB were obtained by centrifugation for 15 min at 10,000 r min^−1^ and sterilized through 0.22 μm filter. Prior to antioxidant and antibacterial experiments, CFS of each LAB strain was adjusted to pH 7.0.

### 2.5. Exopolysaccharide Production

Exopolysaccharide (EPS) of LAB isolates was extracted and quantified as described previously [15]. The LAB were incubated in MRS medium supplemented with 2% sucrose for 36 h at 37 °C to obtain CFS. Then 9.0 mL of CFS was mixed with 1.0 mL of 40% trichloroacetic acid and stood at 4 °C for 12 h to precipitate and remove proteins by centrifugation with 10,000 r min^−1^ for 20 min at 4 °C. The supernatant was transferred into a new tube. Then 3-fold volume of aqueous ethanol was added to left overnight and centrifuged to collect the precipitate. Finally, the precipitation was reconstituted in water, dialyzed by a dialysis bag (MW = 1000 Da) for 24 h, and freeze-dried using a vacuum freeze-dryer (ScientZ-10N, Xinzhi Biotechnology Co., Ltd., Ningbo, China) to obtain crude EPS. Using glucose as a standard, the EPS concentration was measured by the colorimetric phenol-sulfuric acid assay as previously reported [3]. The absorbance was determined at 490 nm by a multimode microplate reader (SpectraMax M2, Molecular Devices, San Jose, CA, USA). The equation for measuring the standard curve was y = 0.0039x + 0.0033 (R^2^ = 0.9994).

### 2.6. Organic Acid Production

The organic acid production in CFS was assessed using titration, with 0.1 mol L^−1^ NaOH and phenolphthalein as an indicator, and expressed as g L^−1^ lactic acid equivalent [17].

### 2.7. Antioxidant Activity Assay

#### 2.7.1. DPPH Radical Scavenging Activity

The DPPH radical scavenging activity was evaluated as described earlier [18]. Briefly, 1.0 mL of CFS and 0.2 mmol L^−1^ ethanolic DPPH radical were mixed with equal volumes. After incubation at 25 °C for 30 min in the dark, the absorbance of the mixture was measured at 517 nm by a spectrophotometer (Specord 200, Analytik Jena, Jena, Germany). The blank was carried out with anhydrous ethanol instead of DPPH solution, whilst the blank medium instead of CFS was used as the control. The DPPH radical scavenging rate was calculated as an inhibition percentage according to the following formula: DPPH scavenging activity (%) = [1 − (A_s_ − A_b_)/A_c_] × 100, where A_s_, A_b_, and A_c_ represented the absorbance of the sample group, blank group, and control group, respectively.

#### 2.7.2. ABTS^+^ Radical Scavenging Activity

ABTS^+^ radical scavenging assay was applied using the procedure previously described [18]. Briefly, 7.0 mmol L^−1^ ABTS solution were added to equal volumes of 2.45 mmol L^−1^ potassium persulfate solution and diluted to an absorbance at 734 nm approximately 0.70 to obtain ABTS^+^ working solution. Then 1.0 mL of CFS and ABTS^+^ working solution was mixed with equal volumes. After incubation at 25 °C for 20 min in the dark, the absorbance of the mixture was measured at 734 nm by a spectrophotometer (Specord 200, Analytik Jena, Jena, Germany). The ABTS^+^ radical scavenging rate was calculated as an inhibition percentage according to the following formula: ABTS^+^ scavenging activity (%) = [1 − (A_s_ − A_b_)/A_c_] × 100, where A_s_, A_b_, and A_c_ represented the absorbance of the sample group, blank group, and control group, respectively.

#### 2.7.3. Hydroxyl Radical (·OH) Scavenging Activity

The ·OH radical scavenging assay was applied using the procedure previously described with slight modifications [18]. Briefly, 0.6 mL of 2.0 mmol L^−1^ FeSO_4_, 6.0 mmol L^−1^ salicylic acid, 2.0 mmol L^−1^ H_2_O_2_, and CFS of LAB were mixed in sequence at volume ratio 3:3:3:1. After incubation at 37 °C for 30 min in the dark, the absorbance of the mixture was measured at 562 nm by a spectrophotometer (Specord 200, Analytik Jena, Jena, Germany). The ·OH radical scavenging rate was calculated as an inhibition percentage according to the following formula: ·OH scavenging activity (%) = [1 − (A_s_ − A_b_)/A_c_] × 100, where A_s_, A_b_, and A_c_ represented the absorbance of the sample group, blank group, and control group, respectively.

#### 2.7.4. Superoxide Anion Radical (·O_2_^−^) Scavenging Activity

·O_2_^−^ radical scavenging activity was measured as described earlier with minor modifications [19]. Briefly, 100 μL of CFS of LAB was added to 1.8 mL of 50 mM Tris-HCl solution (pH 8.2) and was placed in a water bath at 25 °C for 10 min. Then 100 μL of 10 mM pyrogallol solution was added. After incubation at 25 °C for 4 min in the dark, the absorbance of the mixture was measured at 325 nm by a spectrophotometer (Specord 200, Analytik Jena, Jena, Germany). The superoxide anion radical scavenging rate was calculated as an inhibition percentage according to the following formula: ·O_2_^−^ scavenging activity (%) = [1 − (A_s_ − A_b_)/A_c_] × 100, where A_s_, A_b_, and A_c_ represented the absorbance of the sample group, blank group, and control group, respectively.

#### 2.7.5. Ferrous Ion Chelating Capacity (FICC)

The FICC was measured as described earlier with minor modifications [19]. Briefly, 100 μL of 2.0 mmol L^−1^ FeSO_4_ solution was added to 1.0 mL of CFS of LAB. After mixing, 100 μL of 5 mM ferrozine solution was added. After incubation at 25 °C for 10 min in the dark, the absorbance of the mixture was measured at 562 nm by a spectrophotometer (Specord 200, Analytik Jena, Jena, Germany). The ferrous ion chelating rate was calculated as an inhibition percentage according to the following formula: FICC (%) = [1 − (A_s_ − A_b_)/A_c_] × 100, where A_s_, A_b_, and A_c_ represented the absorbance of the sample group, blank group, and control group, respectively.

#### 2.7.6. Ferric Reducing Antioxidant Power (FRAP)

FRAP assay was determined using the method described by Shori et al. with slight modification [20]. Briefly, 10 mmol L^−1^ TPTZ solution in 20 mmol L^−1^ ferric chloride, 40 mmol L^−1^ HCl, and 300 mmol L^−1^ acetate buffer were mixed at volume ratio 1:1:10 to obtain FRAP working solution. Then 0.5 mL of CFS of LAB and FRAP working solution were mixed at volume ratio 1:6. After incubation at 37 °C for 15 min in the dark, the absorbance of the mixture was measured at 593 nm by a spectrophotometer (Specord 200, Analytik Jena, Jena, Germany). FeSO_4_ was used as the standard, and FRAP values of samples were expressed as FeSO_4_ equivalent (μmol L^−1^).

### 2.8. Antibacterial Assay

The antibacterial activity of LAB strains was evaluated according to agar well diffusion methods with slight modifications [11]. The indicator bacteria were grown with shaking (180 r min^−1^) for 16 h at 37 °C, and each culture broth was adjusted to an absorbance of 0.2 at 600 nm. Next 100 µL of pathogenic cultures was spread onto LB plates, and wells of 9 mm diameter were made in the plate. Then 50 µL of CFS was added into each well and the diameters of inhibition zone (mm) were measured after 24 h incubation at 37 °C.

### 2.9. Antioxidant Effect of Fermented Goat Milk

Functional fermented goat milk was prepared as previously described [21]. For each LAB strain, 100 mL of fresh goat milk was heated for 10 min at 95 °C and lest to cool naturally for 60 min. Then 1.0 mL of LAB cells (5 × 10^8^ CFU mL^−1^) were added into 100 mL of the cooled milk to ferment for 20 h at 37 °C. All the samples reached a pH value of 4.2. Afterward, the fermented milks were stored for 24 h at 4 °C in order to perform post-fermenting. Finally, 10.0 mL of each sample was taken, adjusted to the pH 7.0, and determined the antioxidant activity. Simultaneously, goat milk without the addition of LAB was performed as the control.

### 2.10. Statistical Analysis

All assays were carried out in triplicate in this study. Statistical comparisons, principal component analysis (PCA), and entropy weight improved technique for order preference by similarity to an ideal solution (EW-TOPSIS) analysis were performed with SPSS Statistics 20 and Origin Pro 2021. Data were expressed as mean ± standard deviation, and the significance of the received data was assessed using one-way analysis of variance (ANOVA) (*p* < 0.05).

## 3. Results

### 3.1. Isolation of LAB Strains and Identification by 16S rRNA Sequencing

A total of 80 strains were screened from rubing cheese to produce calcium-dissolving circles on MRS plates. Out of this, seven selected isolates with the typical features of LAB were further identified using 16S rRNA gene sequence analysis. As presented in Figure 1, phylogenetic tree analysis of the isolates showed that the seven strains belonged to different species including *Lactococcus lactis* (W3A and W3C), *Streptococcus lutetiensis* (W3B), *Enterococcus durans* (W3D), *Leuconostoc mesenteroides* (W3E), *Enterococcus lactis* (W3F), and *Leuconostoc lactis* (W3J).

### 3.2. Safety Evaluation Through Hemolytic Activity and Antibiotic Susceptibility

According to Figure 2, none of the seven LAB strains had obvious hemolytic circles on blood agar plates, suggesting that they had no active hemolytic activity. As shown in Figure 3, the antibiotic susceptibility of 7 strains against 13 tested antibiotics was evaluated. All strains were sensitive to most antibiotics, such as penicillin, piperacillin, ampicillin, cefazolin, amikacin, tetracycline, minocycline, and doxycycline. While strains *S. lutetiensis* W3B, *Leu. mesenteroides* W3E, and *Leu. lactis* W3J were resistant to cephalexin, *E. durans* W3D and W3F were resistant to gentamicin, and *Lac. lactis* W3A, *Lac. Lactis* W3C, and *Leu. mesenteroides* W3E were resistant to vancomycin.

### 3.3. EPS and Organic Acid Production

As shown in Figure 4, all LAB strains were able to produce EPS and organic acid with significant differences. *E. lactis* W3F attained the highest polysaccharide yield (1097.10 mg L^−1^), and other strains attained between 330.57 and 704.04 mg L^−1^ (Figure 4A). *S. lutetiensis* W3B and *Leu. lactis* W3J attained the highest values of titratable acidity (73.38 and 76.68 g L^−1^ lactic acid equivalent, respectively), and other strains attained between 31.83 and 65.43 g L^−1^ lactic acid equivalent (Figure 4B).

### 3.4. Antioxidant Activity

As shown in Table 1, six methods including DPPH, ABTS^+^, ·OH, ·O_2_^−^ free radical scavenging assays, and the FICC and FRAP tests were applied to estimate the antioxidant capacity of LAB strains. All tested strains exhibited strong radical scavenging activity, and the ABTS^+^ and ·OH radical scavenging rates were higher than DPPH and ·O_2_^−^ radical scavenging rates. The ABTS^+^ radical scavenging rates were all over 95%, followed by ·OH radical scavenging rates (86.06–95.26%) and DPPH radical scavenging rates (37.72–53.13%), while ·O_2_^−^ radical scavenging rates were relatively low ranging from 11.88% to 31.30%. Moreover, seven LAB strains had ferrous ion chelating activity, ranging from 23.98% to 54.70%. Notably, there was a relatively big difference between the total reducing power by FRAP analysis in different strains (*p* < 0.05). The total reducing power of *Lac. lactis* W3C and *E. durans* W3D were significantly higher than others, at 266.83 and 273.14 μmol L^−1^ FeSO_4_ equivalent, respectively. The FRAP value of *S. lutetiensis* W3B was the lowest, only 35.56 μmol L^−1^, and the remaining four strains were between 65.11 and 141.53 μmol L^−1^.

### 3.5. Antibacterial Activity

According to Figure 5, all LAB strains exhibited antibacterial effects on 13 common foodborne pathogens, especially *L. monocytogenes*, *S. flexneri*, *S. pyogenes*. Additionally, strains *S. lutetiensis* W3B, *E. durans* W3D, and *Leu. lactis* W3J displayed ≥16.0 mm inhibition zones against at least 10 tested bacteria, whereas W3E showed weak antibacterial activity against all tested strains (inhibitory zones ≤ 16.0 mm).

### 3.6. PCA and EW-TOPSIS

PCA based on characteristic indicators can rapidly distinguish and screen potential functional LAB strains [22]. PCA plot in Figure 6 revealed the contribution and correlation between the 7 strains and 10 various characteristic phenotypes. The two principal components (PC1 and PC2) explained 62.1% of all the variables. These strains were separated along PC1 (39.1% variance) and PC2 (23.0% variance), and *Leu. lactis* W3J and *E. lactis* W3F located in quadrant 1. Generally, a significant correlation was found between the strains in quadrant 1 and their variables [23]. *E. lactis* W3F and *Leu. lactis* W3J located in quadrant 1 was significantly associated with screening index in quadrant 1. It could be inferred that *E. lactis* W3F and *Leu. lactis* W3J located in quadrant 1 had the superior overall comprehensive performance.

EW-TOPSIS method is also an effective evaluation method that accurately and objectively quantifies multiple indicators [24]. As shown in Table 2, the entropy–weight method was applied to calculate weights of the ten items. The model with the highest weight was FICC (17.29%), while DPPH radical scavenging activity had the lowest weight (5.76%). Table 3 shows the ranking results of seven LAB strains based on EW-TOPSIS. This result was consistent with the PCA results, which indicated that the best score was *Leu. lactis* W3J and the second best score was *E. lactis* W3F.

### 3.7. Antioxidant Abilities of Fermented Goat Milk

Figure 7 shows the antioxidant effects of fermented goat milk with *E. lactis* W3F and *Leu. lactis* W3J. Compared with the control group, the fermentation group with LAB significantly enhanced the ·OH, DPPH, ABTS^+^, and ·O_2_^−^ scavenging capacities, ferrous ion chelating activity, and ferric ions reducing power of goat milk. The FRAP value of fermented goat milk with W3F and W3J (295.26 and 336.94 μmol L^−1^ FeSO_4_ equivalent, respectively) was higher than non-fermented goat milk (253.27 μmol L^−1^) (*p* < 0.05). Based on the results, the fermented goat milk displayed strong antioxidant properties. Furthermore, changes of antioxidant activity of goat milk before and after LAB fermentation were highlights probiotic LAB strains suitable for antioxidant foods.

## 4. Discussion

Fermented dairy products provide an abundance of lactic acid bacteria. *Lactococcus*, *Lactobacillus*, and *Leuconostoc* were frequently detected from the fermented dairy product and contributed to the formation of special flavors in the final product [25,26]. In this work, six different species of LAB were isolated from a traditional Chinese rubing cheese, and *Lactococcus*, *Leuconostoc*, and *Enterococcus* were reported in a previous study [13]. *S. lutetiensis*, *L. mesenteroides*, and *E. lactis* were isolated from this cheese for the first time and cultured to determine their abilities. The strain differences were likely due to the choice of medium, culture conditions, and screening criteria. In addition, other factors including raw material, fermentation time and temperature, and external equipment might also influence microbial composition.

Even though LAB are classified as GRAS bacteria, it is necessary to guarantee the safety of the strains. Hemolytic activity is a decisive indicator to measure the safe use of probiotics in food. The spread of antibiotic resistance is another major potential safety concern. Once a large number of live LAB enter the intestine of humans and animals together with certain food or food supplements, there also exists a potential risk of antibiotic resistance spreads through horizontal gene transfer between gut bacteria and environmental species. In this work, seven LAB strains had no hemolytic activity and were sensitive to the vast majority of antibiotics. Among them, certain individuals were likely to be more sensitive to only one or two specific antibiotics, such as cephalexin and vancomycin. This result was consistent with that previously reported by Feng et al. and Haghshenas et al. [10,11]. These studies indicated that the cephalexin or vancomycin resistance was an intrinsic trait of some LAB strains, such as *Leuconostoc*, *Lactobacillus*, and *Pediococcus*. Targeted sequencing and conjugative transfer assays of drug-resistance genes further confirmed that resistance genes of LAB were chromosomally encoded and non-transmissible, so these strains could be safely used in the food industry [27,28].

EPS, one of the important metabolites of some LAB strains, are safe and biologically active macromolecules. They not only had antioxidant, antibacterial, and anti-inflammatory activities, but also could interact with other food components to enhance the sensory and rheological characteristics of foods [6]. Although, numerous EPS-producing LAB strains were screened from traditional fermented foods, the production of most strains was <1000 mg L^−1^ under non-optimized conditions, such as LAB strains from Tempoyak (100–850 mg L^−1^) [3] and pickles (56.15–515.48 mg L^−1^), severely limiting the industrial production of ESP [29]. Different strains, medium components (carbon and nitrogen sources), and fermentation conditions (temperature and time) could affect polysaccharide composition and activity, thereby impacting industrial production and applications [30]. In this study, a high EPS-producing strain, *E. lactis* W3F, was obtained, and the yield reached 1097.10 mg L^−1^ without any optimization. In addition, there were three strains (*Lac. lactis* W3A, *E. durans* W3D, and *Leu lactis* W3J) with over 500 mg L^−1^ EPS production. Therefore, these strains laid a good foundation for further EPS yield improvement and industrial scale-up production.

LAB are named for their main attribute in food fermentations that convert carbohydrates into organic acids, mainly lactic acid. The organic acids played an important role in food fermentation. On the one hand, the growth of most spoilage bacteria could be inhibited when the total acid contents increased and the pH decreased, resulting in the safety of fermented food [31]. On the other hand, acids were the precursor for ketones, alcohols, aldehydes, and esters, all of which would promote the formation of unique flavors [32]. In this work, all isolated LAB strains showed high organic acid production (Figure 4), confirming their capability to reinforce their use in fermented food.

Contamination with pathogenic or spoilage bacteria often causes food safety and human health problems [33]. Dairy products are more susceptible to microbial spoilage due to their rich nutrient content. Consumption of contaminated dairy products with *Listeria* sp., *Salmonella* sp., *Bacillus* sp., *E. coli*, and *S. aureus* can lead to fever, nausea, vomiting, diarrhea, and abdominal pain [34]. However, commonly used sterilization techniques are not suitable for fermented milk. For example, heat treatment usually disrupts the thermos-sensitive nutrients and flavor in fermented milk. The exogenous preservatives can control harmful bacteria, but they also reduce the fermented activity of LAB. Thus, the discovery of LAB with inherent antimicrobial properties is necessary to help combat pathogenic bacteria. Our results showed seven LAB strains had broad-spectrum antibacterial activity, and *S. lutetiensis* W3B, *E. durans* W3D, and *Leu. lactis* W3J showed ≥16.0 mm inhibition zones against at least 10 tested bacteria. These data indicated that the use of these strains can ensure greater safety in food.

Oxidation is one of the main causes of food spoilage and the reduction in shelf life. In addition, the loss of nutrition, color, and functionality and the accumulation of undesirable off-flavors and toxic compounds induced by oxidation could potentially endanger the health of consumers [35]. Antioxidants play an important role in delaying or preventing the oxidation of fermented food and contribute to health promotion as dietary supplements. Since LAB do not possess a complete electron transport chain, many LAB strains need to produce more antioxidant enzymes (glutathione S-transferase and superoxide dismutase) or metabolites (lactic acid and exopolysaccharide) to resist the oxidation damage [7,36,37], so these LAB strains with high antioxidant ability have great potential for the development of functional foods.

Most LAB have the ability to scavenge free radicals. In our study, seven LAB strains not only could scavenge various types of free radicals (DPPH, ABTS^+^, ·OH, ·O_2_^−^), but also could chelate metal ions and reduce ferric ions. Notably, there were intra- and interspecies differences in their antioxidant activity using different experimental approaches. For example, *E. durans* W3D had the highest FRAP value and the lowest ·OH radical scavenging rate. The FICC of *Lac. lactis* W3C was the best, while the ·O_2_^−^ scavenging activity was the worst (Table 1). The reason might be that different LAB species and strains produced different types and amounts of antioxidant substances [7]. Moreover, ESP and organic acid were important antioxidant substances synthesized by LAB. Previously, Saravanan et al. reported that EPS from *Leu. lactis* had high radical antioxidant activity, such as DPPH (6.8–74%) and ·OH radical (27.5–97.8%) activity, as well as metal chelating activity (5.8–72.5%) [38]. In this study, significant difference in ESP and organic acid yield between strains and species further provided possible evidence for variation of the antioxidant activity. This result was comparable to Madjirebaye et al. [14] and Khalil et al. [3], indicating that each strain provided varying contributions to the antioxidant potential in different measurement assays.

Since the results of each antioxidant indicator were different, PCA and EW-TOPSIS methods were used to select the best sample. Taking into account practical applications, antibacterial and metabolite characteristics of LAB were also investigated. The top two were *Leu. lactis* W3J and *E. lactis* W3F (Figure 6 and Table 3). Furthermore, LAB have made a significant contribution to the production of fermented foods as functional foods or dietary supplements. Rizzello et al. [39] and Madjirebaye et al. [14] reported that the antioxidant activity of quinoa flour and soymilk was improved by selected LAB fermentation. This work also confirmed the *Leu. lactis* W3J and *E. lactis* W3F as starter cultures to improve the antioxidant capacity of goat milk. The results provided valuable information for the application of LAB in a variety of antioxidant foods.

## 5. Conclusions

In this study, seven LAB strains were isolated and identified from Chinese traditional cheese. These strains were found to be safe for consumers. Moreover, they showed the ability to produce EPS and organic acid. At the same time, all strains exhibited strong antioxidant and antibacterial activities. PCA and EW-TOPSIS results demonstrated that *Leu. lactis* W3J and *E. lactis* W3F possessed a relatively good performance. At last, goat milk fermented with *Leu. lactis* W3J and *E. lactis* W3F displayed a significant enhancement in antioxidant activity. Therefore, these two LAB strains isolated from Chinese traditional cheese could be considered promising candidates as functional merits.

## Figures and Tables

**Figure 1 microorganisms-13-02743-f001:**
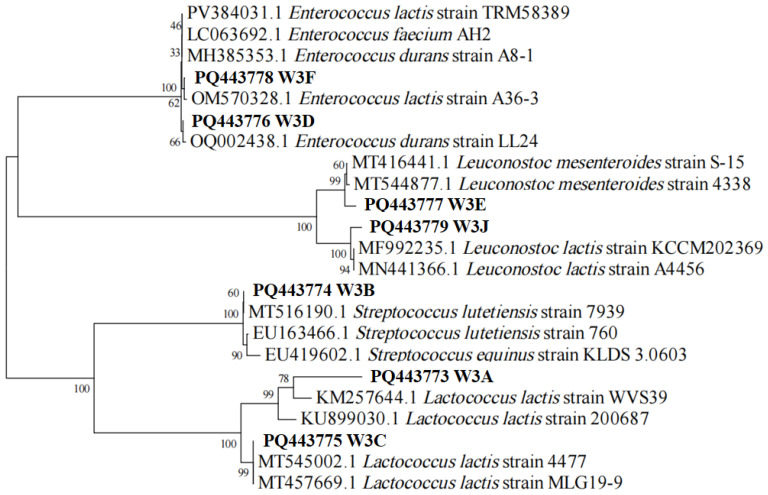
The phylogenetic tree of seven LAB strains based on 16S rRNA sequences. W3A-J indicates strain name. The accession number in the GenBank for the 16S rRNA gene sequence of each strain is shown before the strain name. Numbers on the branches refer to bootstrap values. Bar (0.02) represents sequence divergence.

**Figure 2 microorganisms-13-02743-f002:**
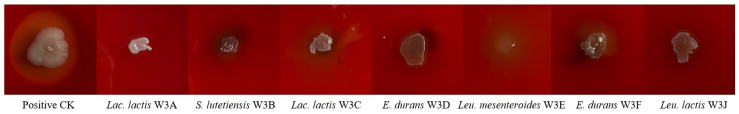
Hemolytic activity of seven LAB strains.

**Figure 3 microorganisms-13-02743-f003:**
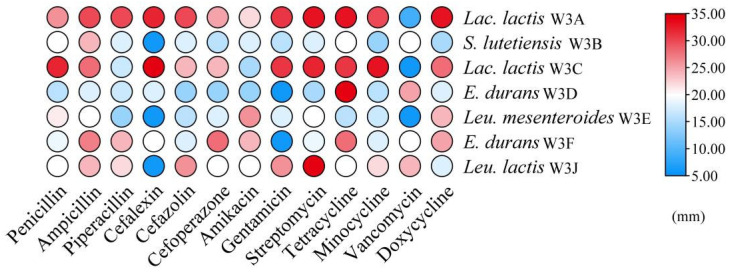
Heat map of antibiotic susceptibility of seve LAB strains. The color scale (blue–red) indicates the diameters of inhibition zone.

**Figure 4 microorganisms-13-02743-f004:**
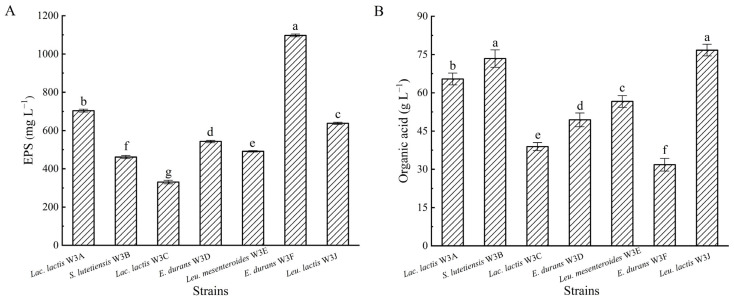
Exopolysaccharide (**A**) and organic acid (**B**) production of seven LAB strains. Different letters above bars represent significant differences (*p* < 0.05).

**Figure 5 microorganisms-13-02743-f005:**
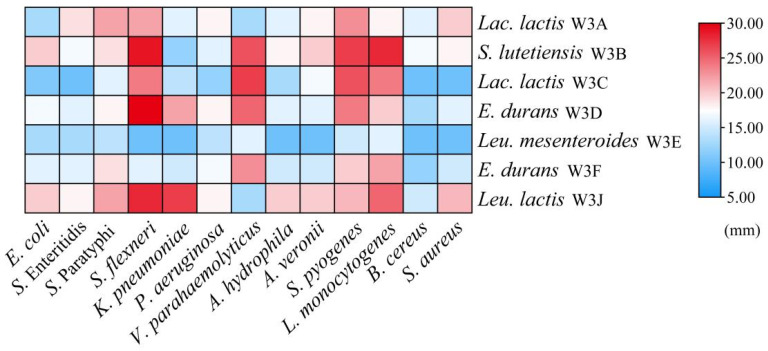
Heat map of antibacterial activities of seven LAB strains. The color scale (blue–red) indicates the diameters of inhibition zone.

**Figure 6 microorganisms-13-02743-f006:**
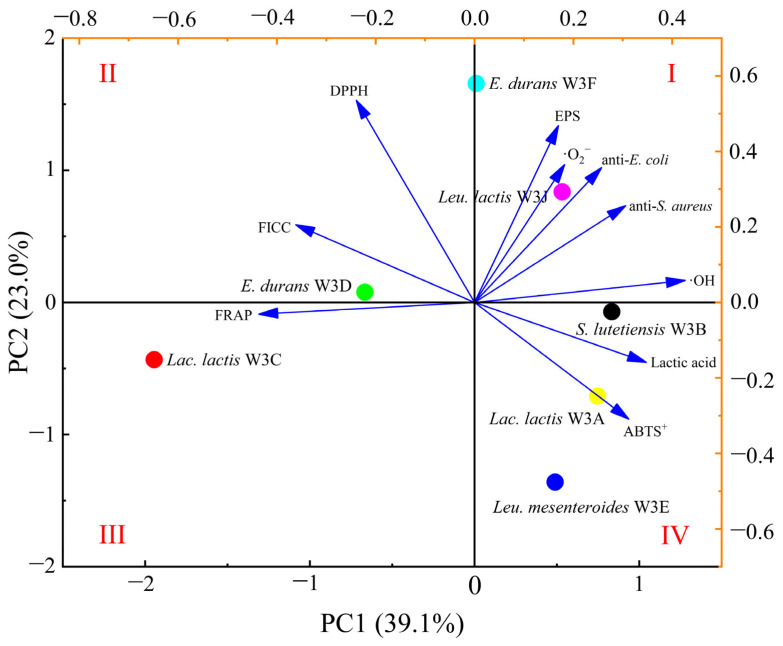
Two-dimensional principal component analysis (PCA) plot based on characteristic phenotypes for selection of LAB strains. The quadrants are labelled with Roman numerals.

**Figure 7 microorganisms-13-02743-f007:**
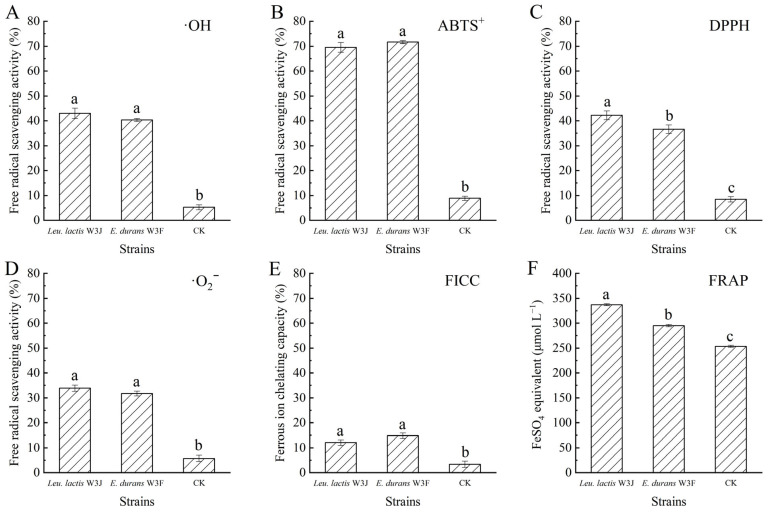
The antioxidant effect of fermented goat milk. (**A**) ·OH radical scavenging activity; (**B**) ABTS^+^ radical scavenging activity; (**C**) DPPH radical scavenging activity; (**D**) ·O_2_^−^ radical scavenging activity; (**E**) Ferrous ion chelating capacity (FICC); (**F**) ferric ion reducing power (FRAP). Different letters above bars represent significant differences (*p* < 0.05).

**Table 1 microorganisms-13-02743-t001:** Antioxidant activity of seven LAB strains.

Strains	·OH	ABTS+	DPPH	·O_2_^−^	FICC (%)	FRAP (μmol L^−1^)
	Radical Scavenging Activity (%)					
*Lac. lactis* W3A	92.69 ± 0.96 ^c^	98.67 ± 0.75 ^a^	44.78 ± 1.43 ^c^	15.99 ± 1.17 ^d^	28.49 ± 1.17 ^c^	65.11 ± 0.76 ^f^
*S. lutetiensis* W3B	95.26 ± 0.38 ^a^	96.43 ± 0.14 ^bc^	46.26 ± 0.60 ^c^	17.24 ± 1.23 ^d^	25.15 ± 0.95 ^d^	35.56 ± 1.42 ^g^
*Lac. lactis* W3C	87.56 ± 0.22 ^d^	95.86 ± 0.30 ^c^	51.76 ± 0.38 ^ab^	11.88 ± 1.27 ^e^	54.70 ± 1.24 ^a^	266.83 ± 1.57 ^b^
*E. durans* W3D	86.06 ± 0.09 ^e^	96.81 ± 0.29 ^b^	50.64 ± 1.35 ^b^	20.26 ± 1.40 ^c^	25.78 ± 1.16 ^d^	273.14 ± 1.36 ^a^
*Leu. mesenteroides* W3E	93.94 ± 0.19 ^b^	98.09 ± 0.53 ^a^	37.72 ± 0.58 ^d^	24.41 ± 1.33 ^b^	23.98 ± 1.10 ^d^	98.66 ± 0.17 ^d^
*E. durans* W3F	93.05 ± 0.03 ^c^	96.67 ± 0.34 ^b^	53.13 ± 0.36 ^a^	31.30 ± 0.41 ^a^	38.81 ± 1.60 ^b^	79.53 ± 0.03 ^e^
*Leu. lactis* W3J	94.44 ± 0.08 ^b^	96.94 ± 0.65 ^b^	52.13 ± 1.45 ^ab^	22.00 ± 1.06 ^c^	39.44 ± 1.38 ^b^	141.53 ± 0.62 ^c^

All values are means ± standard deviation of three independent experiments. Different superscript letters within each column represent significant differences (*p* < 0.05).

**Table 2 microorganisms-13-02743-t002:** Weight distribution of each indicator using the entropy–weight method.

Index	Ej	Dj	Wj
·OH	0.88	0.12	7.41%
ABTS^+^	0.84	0.16	9.46%
DPPH	0.90	0.10	5.76%
FRAP	0.79	0.21	12.43%
FICC	0.71	0.29	17.29%
·O2^−^	0.86	0.14	8.41%
EPS	0.83	0.17	10.35%
Lactic acid	0.86	0.14	8.46%
Anti-*S. aureus*	0.81	0.19	11.33%
Anti-*E. coli*	0.85	0.15	9.12%

Ej represents the information entropy value, Dj represents information utility value, and Wj represents the comprehensive weight of indicators.

**Table 3 microorganisms-13-02743-t003:** The ranking results of seven LAB strains based on the EW-TOPSIS method.

Strains	D+	D−	Ci	Ranking
*Lac. lactis* W3A	0.592	0.478	0.447	4
*S. lutetiensis* W3B	0.682	0.44	0.392	6
*Lac. lactis* W3C	0.662	0.58	0.467	3
*E. durans* W3D	0.619	0.444	0.417	5
*Leu. mesenteroides* W3E	0.706	0.361	0.338	7
*E. durans* W3F	0.496	0.558	0.53	2
*Leu. lactis* W3J	0.406	0.597	0.595	1

D+ is the distance between each evaluation index and the positive ideal solution, D− refers to the distance between each evaluation index and the negative ideal solution, and Ci represents the relative proximity.

## Data Availability

The raw data supporting the conclusions of this article will be made available by the authors on request.

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
