# Peer review of "Biological Activities of Lactic Acid Bacteria Isolated from Chinese Traditional Cheese and the Application in Antioxidant Foods"

_microorganisms, 2025, doi:10.3390/microorganisms13122743_

Round 1

Reviewer 1 Report

Comments and Suggestions for Authors

The manuscript entitled “Biological Activities of Lactic Acid Bacteria Isolated from Chinese Traditional Cheese and the Application in Antioxidant Foods” deals with the isolation, characterization, and selection of lactic acid bacteria (LAB) from traditional rubing cheese, with emphasis on antioxidant and antibacterial activities and their application in fermented goat milk. The authors have shown that seven LAB strains present a combination of safety, metabolite production (EPS and organic acids), strong radical-scavenging activity, and broad-spectrum antimicrobial effects. They further applied PCA and EW-TOPSIS to identify the most promising strains (Leu. lactis W3J and E. lactis W3F), which were then used to ferment goat milk, improving its antioxidant potential. These results are interesting because they contribute to identifying novel indigenous LAB strains with potential use as starter cultures for antioxidant functional foods. The manuscript is generally well written, and conclusions are supported by data. Some points should be addressed to improve clarity, methodological rigor, and overall scientific quality.

Introduction

The introduction section is good, but the novelty of isolating these specific strains from rubing cheese should be more clearly emphasized. Although the authors mention some studies with isolated strains from fermented food, the introduction would benefit from clear articulation of the knowledge gap.

Methodology

Section 2.3) LAB were cultivated together or separated? What is CFS?

Several methods require additional information to ensure reproducibility:

Correct minor inconsistencies in unit notation (e.g., “r min-1” vs. “rpm”).

Exopolysaccharide quantification: the authors should specify whether EPS was dialyzed or deproteinized before quantification. Phenol-sulfuric assay is sensitive to interfering compounds, and clarification is needed.

Organic acid determination: the manuscript states that titratable acidity was expressed as lactic acid equivalent, but the exact titration endpoint (pH value or indicator) is not mentioned.

Antioxidant assays: all radical-scavenging methods reference previous studies, but important parameters (e.g., reaction volumes, calibration curves, controls) should be explicitly stated.

Antibacterial assay: the authors do not specify whether the CFS were neutralized or if pH effects were controlled. This is essential to distinguish between true antimicrobial compounds and acidity.

Results

Table 1 presents numerous antioxidant assays, but variability is high. The text should include a brief discussion explaining biological sources of intra-species variability.

Figures 2–5 are clear, but some captions lack critical details (e.g., strain names should be repeated for clarity).

Table 3 should indicate the meaning of D+, D–, and Ci directly in the caption.

The authors state that resistance to vancomycin and cephalexin is intrinsic. However, the discussion should explicitly mention whether genomic data or literature was used to confirm that these traits are non-transmissible, as this is critical for safety assessment.

The fermentation of goat milk with selected strains is interesting, but the authors should provide:

pH and acidity of fermented milk, since these factors influence antioxidant assays.

A clearer explanation of the mechanism by which LAB fermentation enhances antioxidant activity.

The discussion is well structured but could better integrate references when comparing antioxidant performance to previous studies.

Consider adding a short paragraph on potential industrial relevance and limitations (e.g., scalability of EPS production).

Language and formatting

Minor English corrections are needed throughout.

Ensure consistency in species nomenclature (italicization, abbreviations).

Reviewer 2 Report

Comments and Suggestions for Authors

The innovative nature of the research described in this manuscript lies in the isolation and identification of LAB strains from traditional Chinese cheese, the examination of selected properties of these strains with a strong emphasis on antioxidant activity, and the additional assessment of the antioxidant activity of goat milk fermented with two of the seven isolates. The research is very interesting. However, section 2, Materials and Methods, in particular, requires significant improvement.

Detailed comments:

Lines 11-12: The term "the probiotic properties of LAB" is too general and inappropriate for this manuscript. This should be more specific, as the isolates' antioxidant and antibacterial properties, as well as antibiotic resistance, were studied.

Lines 57-59: In my opinion, calling lactic acid bacteria (LAB) a substitute for chemical preservation using lactic acid is overstated. The substitute may be another organic acid, e.g. citric acid. These methods belong to two different groups of food preservation methods: chemical and biological.

Lines 75-191, Section 2. Materials and Methods: There is no description of the materials used in the studies. A note concerns the origin of the microbiological media and numerous chemical reagents used in all the methods described in this section. There is no description of the biological material, i.e., the strains used, in Section 2.6. Their origins are not described.

Line 80: Was the (mMRS) with or without agar? Was the incubation aerobic or anaerobic?

Lines 95-104, section 2.2.2. Antibiotic Susceptibility: What is the origin of the antibiotic discs?

Lines 108, 114, and following lines: What does the abbreviation "CFS" stand for? This should be explained in the first sentence where it is used.

Lines 108-109: What is the origin of trichloroacetic acid?

Lines 111-112: An abbreviation for the EPS concentration was used: "Using glucose as a standard, the EPS concentration was measured by the phenol-sulfuric acid assay" – this is unclear.

Line 119: Unclear wording: "erhanolic DPPH radical solution."

Lines 121, 131, 141, 150, 159, 169: What type of spectrophotometry device was used?

Lines 180-181 and 259 (Fig.5): The Latin names of the bacteria should be written as follows: Salmonella Paratyphi and Salmonella Enteritidis, because these are abbreviations of Salmonella enterica subsp. enterica ser. Paratyphi and Salmonella enterica subsp. enterica ser. Enteritidis.

Line 181: There is no description of the preparation of the bacterial cultures for the experiment. On what medium were the bacteria grown and under what conditions?

Line 188: Was this a single LAB isolate or several isolates obtained from rubing cheese? The description is unclear.

Line 189: How many fermented milks were there? The description is unclear.

Line 209, Figure 1: Were the isolate strains named W3A-J? This wasn't explained in the figure title. Also, there are no numbers in brackets, as indicated in the title of the figure.

Line 274, Figure 6: What do the arrows with E. coli and S. aureus in quadrant 1 mean?
